# DecAF: Joint Decoding of Answers and Logical Forms for Question Answering over Knowledge Bases

**Donghan Yu**[1]*, **Sheng Zhang**[2], **Patrick Ng**[2], **Henghui Zhu**[2], **Alexander Hanbo Li**[2],
**Jun Wang**[2], **Yiqun Hu**[2], **William Wang**[2], **Zhiguo Wang**[2], **Bing Xiang**[2]
[1]Carnegie Mellon University  [2]AWS AI Labs, New York, USA
[1]dyu2@cs.cmu.edu; [2]{zshe, patricng, zhiguow}@amazon.com

## Abstract

Question answering over knowledge bases (KBs) aims to answer natural language questions with factual information such as entities and relations in KBs. Previous methods either generate logical forms that can be executed over KBs to obtain final answers or predict answers directly. Empirical results show that the former often produces more accurate answers, but it suffers from non-execution issues due to potential syntactic and semantic errors in the generated logical forms. In this work, we propose a novel framework DecAF that jointly generates both logical forms and direct answers, and then combines the merits of them to get the final answers. Moreover, different from most of the previous methods, DecAF is based on simple free-text retrieval without relying on any entity linking tools — this simplification eases its adaptation to different datasets. DecAF achieves new state-of-the-art accuracy on WebQSP, FreebaseQA, and GrailQA benchmarks, while getting competitive results on the ComplexWebQuestions benchmark.[1]

## 1 Introduction

Knowledge Bases Question Answering (KBQA) aims to answer natural language questions based on knowledge from KBs such as DBpedia (Auer et al., 2007), Freebase (Bollacker et al., 2008) or Wikidata (Vrandečić & Krötzsch, 2014). Existing methods can be divided into two categories. One category is based on semantic parsing, where models first parse the input question into a logical form (e.g., SPARQL (hommeaux, 2011) or S-expression (Gu et al., 2021)) then execute the logical form against knowledge bases to obtain the final answers (Das et al., 2021; Gu et al., 2021; Ye et al., 2022). The other category of methods directly outputs answers without relying on the the logical-form executor (Lan et al., 2019; Sun et al., 2019; Saxena et al., 2022; Oğuz et al., 2022). They either classify the entities in KB to decide which are the answers (Sun et al., 2019) or generate the answers using a sequence-to-sequence framework (Saxena et al., 2022; Oğuz et al., 2022).

Previous empirical results (Ye et al., 2022; Das et al., 2021; Gu et al., 2022) show that the semantic parsing based methods can produce more accurate answers over benchmark datasets. However, due to the syntax and semantic restrictions, the output logical forms can often be non-executable and thus would not produce any answers. On the other hand, direct-answer-prediction methods can guarantee to generate output answers, albeit their answer accuracy is usually not as good as semantic parsing based methods, especially over complex questions which require multi-hop reasoning (Talmor & Berant, 2018). To our knowledge, none of the previous studies have leveraged the advantages of both types of methods. Moreover, since knowledge bases are usually large-scale with millions of entities, most previous methods rely on entity linking to select relevant information from KB for answering questions. However, these entity linking methods are usually designed for specific datasets, which inevitably limits the generalization ability of these methods.

In this paper, we propose a novel framework DecAF to overcome these limitations: (1) Instead of relying on only either logical forms or direct answers, DecAF jointly decodes them together, and further combines the answers executed using logical forms and directly generated ones to obtain

---

*Work done during internship at AWS AI Labs

[1]Our code is available at https://github.com/awslabs/decode-answer-logical-form

the final answers. Thus the advantages of both methods can be leveraged in our model. Moreover, unlike previous methods using constrained decoding (Chen et al., 2021a) or post revision (Das et al., 2021) to produce more faithful logical forms, we simply treat logical forms as regular text strings just like answers during generation, reducing efforts of hand-crafted engineering. (2) Different from previous methods which rely on entity linking (Yih et al., 2015; Li et al., 2020) to locate entities appeared in questions and then retrieve relevant information from KB, DECAF linearizes KBs into text documents and leverages free-text retrieval methods to locate relevant sub-graphs. Based on this simplification, DECAF brings better adaptation to different datasets and potentially different KBs due to the universal characteristic of text-based retrieval. Experiments show that simple BM25 retrieval brings surprisingly good performance across multiple datasets.

We conduct experiments on four benchmark datasets including WebQSP (Yih et al., 2016), ComplexWebQuestions (Talmor & Berant, 2018), FreebaseQA (Jiang et al., 2019), and GrailQA (Gu et al., 2021). Experiment results show that our model achieves new state-of-the-art results on WebQSP, FreebaseQA, and GrailQA benchmarks, and gets very competitive results on the ComplexWebQuestions benchmark. This demonstrates the effectiveness of DECAF across different datasets and question categories.

## 2 RELATED WORK

**Semantic parsing based methods for KBQA** first parse the input question into a logical form (LF) then execute it against KB to obtain the answers. ReTrack (Chen et al., 2021a) uses a grammar-based decoder to generate LFs based on pre-defined grammar rules, and a semantic checker to discourage generating programs that are semantically inconsistent with KB. RnG-KBQA (Ye et al., 2022) first enumerates possible LFs based on the entity in the input question. Then a ranking-and-generation framework is applied to output the final LF. ArcaneQA (Gu & Su, 2022) generates the LFs dynamically based on the execution results of LFs generated at intermediate steps. TIARA (Shu et al., 2022) proposes a multi-grained retrieval method to select relevant KB context for logical form generation. All these methods rely on an external executor such as a SPARQL server to execute LFs for final answers. If the LFs are not executable, then no answers will be produced.

**Direct-Answer-Prediction methods for KBQA** directly output answers without relying on the LF executor. PullNet (Sun et al., 2019) retrieves a subgraph of KB related to the input question and applies graph neural networks to predict the answer entities in the subgraphs. KGT5 (Saxena et al., 2022) uses a sequence-to-sequence framework to directly generate answers only based on the input question. UniK-QA (Oğuz et al., 2022) is also based on a sequence-to-sequence framework, but it first retrieves relevant triplets from KB and then generates answers based on the combination of the input question and retrieved triplets. Although answers can always be produced without the need of LF executor, this type of method usually underperforms semantic parsing based methods on public benchmarks (Talmor & Berant, 2018; Gu et al., 2021; 2022).

**Entity Linking & Knowledge Linearization.** Real-world KBs are usually very large and with millions of entities and triplets. The algorithm to ground the input question onto a relevant sub-graph of KB is important. Entity linking is the most common way for this. CBR-KBQA (Das et al., 2021) combines an off-the-shelf model ELQ (Li et al., 2020) with an NER system provided by Google Cloud API for entity linking. RnG-KBQA (Ye et al., 2022) also uses ELQ for the WebQSP dataset, while it uses a BERT-based (Devlin et al., 2019) NER system and trains another BERT-based entity disambiguation model for the GrailQA dataset. Previous works usually design different methods to optimize the performance of different datasets. A recent study Soliman et al. (2022) also shows that entity linking models are usually domain-specific and hard to transfer across domains. Different from these methods, DECAF reduces this burden by linearizing KBs to text documents and leveraging simple text-retrieval methods. Experimental results show that this is not only more general but also empirically effective. Similar to our method, UniK-QA (Oğuz et al., 2022) also linearizes the KB and conducts retrieval. However, UniK-QA still requires entity linking (Yih et al., 2015) to reduce the retrieval range, and it only generates direct answers without studying logical forms for questions requiring complex reasoning. Besides the studies above, UnifiedSKG (Xie et al., 2022) is relevant since it studies the generation of logical forms and direct answers for KBQA. However, it does not study combining the advantages of both logical forms and direct answers, and further assumes that the ground-truth question entities are provided, which dramatically eases this

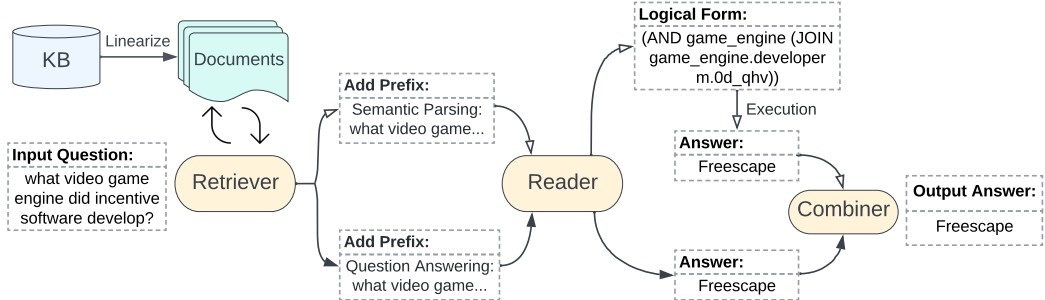

Figure 1: Model framework of DECAF. We use text-based retrieval instead of entity linking to select question-related information from the KB. Then, we add different prefixes into the reader to generate logical forms and direct answers respectively. The logical-form-executed answers and directly-generated answers are combined to obtain the final output.

problem. Li et al. (2021) generates either an answer or an LF for an input question based on model choice, but it studies the problem of Text-Table QA, which is inherently different from KBQA.

## 3 METHOD

In order to (1) leverage the advantages of both logical forms and direct answers and (2) reduce the dependency on entity linking models, we propose a novel framework DECAF. As shown in Figure 1, the whole KB is first transformed into text documents. Then, for an input question, the retriever retrieves relevant passages from linearized KB documents, which will be combined with the input question into the reader. DECAF reader is a sequence-to-sequence generative model and uses different prefixes to generate logical forms (LFs) and direct answers respectively. Finally, the executed answers by LFs and generated direct answers are combined to obtain the final answers.

### 3.1 KNOWLEDGE BASES LINEARIZATION

Given a question, retrieving relevant information from KBs is non-trivial since KBs are large-scale and complicated with both semantic (names of entities and relations) and structural (edge between entities by relations) information. On the other hand, recent studies have shown the effectiveness of text-based retrieval for question answering (Chen et al., 2017; Karpukhin et al., 2020; Izacard & Grave, 2021). By linearizing KBs to text corpus, we can easily leverage these successful text-retrieval methods to select semantically relevant information from KBs.

We describe how to linearize the knowledge base. Considering the original KB format to be the most common Resource Description Format (RDF), which contains triplets composed of one head entity, relation, and tail entity. For example, (*Freescape*, *game_engine.developer*, *Incentive Software*) means that Incentive Software is the developer of a game engine called Freescape. To linearize this triplet, we simply concatenate them with spaces to be a sentence as *Freescape game engine developer Incentive Software.* Note that we also replace punctuation marks in relation to spaces. In this case, this sentence contains both semantic information (entity and relation names) and structural information (relation between two entities). Then we further group sentences with the same head entity to become a document like *Freescape game engine developer Incentive Software. Freescape release date 1987...* This grouping is mainly to preserve the structural information of the 1-hop subgraph corresponding to the head entity. In some KBs like Freebase (Bollacker et al., 2008), there exist hyper-triplets that express complicated relationship, which is discussed in Appendix A.1. Following (Karpukhin et al., 2020), we truncate each document into multiple non-overlap passages with a maximum of 100 words. In the next section, we show how to retrieve from these passages.

### 3.2 RETRIEVAL

The retriever retrieves relevant passages from the linearized KB based on the input question. We consider two kinds of retrieval methods: sparse retrieval and dense retrieval. For sparse retrieval, we

use BM25 (Robertson et al., 2009), which is based on TF-IDF scores of sparse word match between input questions and KB-linearized passages. For dense retrieval, we apply the DPR (Karpukhin et al., 2020) framework, which is based on similarity in the embedding space between input questions and passages from two fine-tuned BERTs (Devlin et al., 2019). We refer readers to the original paper for details of the fine-tuning process. During inference, suppose there are totally $N$ passages in the knowledge source $\{p_1, p_2, \ldots, p_N\}$. DPR applies the passage encoder $E_P(\cdot)$ to encode all the passages and store embeddings in memory. For an input question $q$, DPR applies the question encoder $E_Q(\cdot)$ to obtain its representation, and then the passages are retrieved based on the dot-product similarity: $I_{\text{retrieve}} = \text{argtop-k}_i(E_P(p_i) \cdot E_Q(q))$. Then it applies FAISS(Johnson et al., 2019) to conduct an efficient similarity search due to the large number of passages $N$. Through this step, we can retrieve $|I_{\text{retrieve}}| \ll N$ passages which are potentially relevant to the input question.

## 3.3 READING

The reader takes the retrieved passages and the original question as input and generates the targeted output. Recent advanced sequence-to-sequence models like BART (Lewis et al., 2020) and T5 (Raffel et al., 2020) can be utilized here. In order to answer complicated multi-hop questions, cross-passage reasoning is important for the reader. One way is to concatenate all the passages and let the self-attention in the Transformer module capture these patterns. However, this can be inefficient when the number of retrieved passages is very large because of the quadratic computation complexity in self-attention. To achieve both cross-passage modeling and computation efficiency, we apply Fushion-in-Decoder (FiD) (Izacard & Grave, 2021) based on T5 as the reader model. FiD encodes the concatenation of the input question and each passage separately but decodes the output tokens jointly. Specifically, we denote the question as $q$ and the retrieved passages as $\{p_{r_i}|r_i \in I_{\text{retrieve}}\}$. The encoding process for each passage $p_{r_i}$ is:

$$\mathbf{P}_i = \text{Encoder}(\text{concat}(q, p_{r_i})) \in \mathbb{R}^{L_p \times H}, \tag{1}$$

where $H$ is the hidden dimension, and $L_p$ is the total sequence length of a question concatenated with a passage. T5-Encoder$(\cdot)$ is the encoder module of T5. Then the token embeddings of all passages output from the last layer of the encoder are concatenated and sent to the decoder to generate the output tokens $T_{\text{output}}$:

$$T_{\text{output}} = \text{Decoder}([\mathbf{P}_1; \mathbf{P}_2; \cdots ; \mathbf{P}_{|I_{\text{retrieve}}|}]) \in \mathbb{R}^{|I_{\text{retrieve}}|L_p \times H}) \tag{2}$$

By concatenating the encoder output embeddings, the decoder can generate outputs based on joint modeling of multiple passages.

## 3.4 JOINT DECODING ANSWERS AND LOGICAL FORMS

Motivated by the success of adding prefixes to control the generation of large-scale language models (Raffel et al., 2020; Xie et al., 2022), we use a shared sequence-to-sequence model to generate both logical forms and direct answers, and differentiate the two processes by prompting the model with different prefixes. The encoding-decoding process in Equation (1) and (2) becomes:

$$\mathbf{P}_i^{\text{answer}} = \text{Encoder}(\text{concat}(\text{prefix}^{\text{answer}}, q, p_{r_i})), \ T_{\text{answer}} = \text{Decoder}([\mathbf{P}_1^{\text{answer}}; \cdots ; \mathbf{P}_{|I_{\text{retrieve}}|}^{\text{answer}}]); \tag{3}$$

$$\mathbf{P}_i^{\text{LF}} = \text{Encoder}(\text{concat}(\text{prefix}^{\text{LF}}, q, p_{r_i})), \ T_{\text{LF}} = \text{Decoder}([\mathbf{P}_1^{\text{LF}}; \cdots ; \mathbf{P}_{|I_{\text{retrieve}}|}^{\text{LF}}]) \tag{4}$$

where $T_{\text{answer}}$ and $T_{\text{LF}}$ are the output answer tokens and logical form tokens respectively. prefix$^{\text{answer}}$ and prefix$^{\text{LF}}$ are the prefixes for answer generation and logical form generation respectively, which we set as *Question Answering:* and *Semantic Parsing:*. For example, given the question *What video game engine did incentive software develop?*, we'll first retrieve relevant passages using the retriever. Then we add these two prefixes to produce two different inputs: *Question Answering: what video game engine did incentive software develop?* and *Semantic Parsing: what video game engine did incentive software develop?"* For the first input, we train the model to generate the target output answer *Freescape* directly[2]. While for the second input, we aim to

---

[2]For direct answer generation, we only consider returning one single answer. We'll discuss the possibility of returning multiple answers together in Appendix A.3

| Model | WebQSP | | CWQ | FreebaseQA |
|---|---|---|---|---|
| | Hits@1 | F1 | Hits@1 | Hits@1 |
| PullNet (Sun et al., 2019) | 67.8 | 62.8 | 47.2 | - |
| EmQL (Sun et al., 2020) | 75.5 | - | - | - |
| NSM$_{+h}$ (He et al., 2021) | 74.3 | - | 53.9 | - |
| FILM (Verga et al., 2021) | 54.7 | - | - | 63.3 |
| KGT5 (Saxena et al., 2022) | 56.1 | - | 36.5 | - |
| CBR-SUBG (Das et al., 2022) | 72.1 | - | - | 52.1 |
| SR+NSM (Zhang et al., 2022) | 69.5 | 64.1 | 50.2 | - |
| UniK-QA (Oğuz et al., 2022) | 79.1 | - | - | - |
| QGG (Lan & Jiang, 2020) | - | 74.0 | 44.1 | - |
| HGNet (Chen et al., 2021b) | 70.6 | 70.3 | 65.3 | - |
| ReTrack (Chen et al., 2021a) | 74.7 | 74.6 | - | - |
| CBR-KBQA (Das et al., 2021) | - | 72.8 | **70.4** | - |
| ArcaneQA (Gu & Su, 2022) | - | 75.3 | - | - |
| Program Transfer (Cao et al., 2022) | 74.6 | 76.5 | 58.1 | - |
| RnG-KBQA (Ye et al., 2022) | - | 75.6 | - | - |
| RnG-KBQA (T5-large)$^\star$ | - | 76.2 ± 0.2 | - | - |
| DECAF (BM25 + FiD-large) | 79.0 ± 0.4 | 74.9 ± 0.3 | 68.1 ± 0.5 | 78.8 ± 0.5 |
| DECAF (DPR + FiD-large) | 80.7 ± 0.2 | 77.1 ± 0.2 | 67.0 ± 0.4 | **79.0** ± 0.6 |
| DECAF (BM25 + FiD-3B) | - | - | **70.4** | - |
| DECAF (DPR + FiD-3B) | **82.1** | **78.8** | - | - |

Table 1: Results on the test splits of 3 benchmark datasets: WebQSP, CWQ, and FreebaseQA. The two blocks of baselines are direct-answer-prediction and semantic parsing based methods respectively. We run 5 independent experiments for FiD-large based DECAF and report mean and standard deviation. $^\star$ means that we replace the original reader T5-base with T5-large and rerun experiments to have a fair comparison with our method.

generate the logical form (`AND cvg.computer_game_engine (JOIN cvg.computer_game_-engine.developer m.0d_qhv)`), which is simply treated as text strings during generation without constrained decoding. However, instead of directly generating the entity ID like *m.0d_qhv* in the logical form, we replace it with the corresponding entity name like *Incentive Software* and add special tokens "[" and "]" to identify it. The revised logical form to be generated becomes (`AND cvg.computer_game_engine (JOIN cvg.computer_game_engine.developer [ Incentive Software ])`). After generation, we replace the entity names with their IDs for execution. To ensure the 1-1 mapping between entity name and ID, we preprocess the KB by name disambiguation, which is introduced in Appendix A.2. During training, we fine-tune the whole reader in a multi-task learning manner, where one input question contributes to two training data pairs with one for answer generation and the other one for logical form generation. During inference, we use beam search to generate multiple candidates for both logical forms and answers, with the same beam size $B$.

Next, we show how we combine them to obtain the final answer. We first execute these logical forms against the KB using an executor[3]. Suppose the list of executed answer set is $[A_1^{\mathrm{LF}}, \cdots, A_{B'}^{\mathrm{LF}}]$ ($B' \leq B$ since some logical forms can be non-executable) and the list of directly generated answer set is $[A_1^{\mathrm{answer}}, \cdots, A_B^{\mathrm{answer}}]$. We consider two situations: (1) If $B' = 0$ means none of these logical forms are executable, the final answer is simply $A_1^{\mathrm{answer}}$; (2) Otherwise when $B' \geq 1$, we use weighted linear combination: we first assign the score $\lambda S(k)$ for $A_k^{\mathrm{LF}}$ and the score $(1 - \lambda)S(k)$ for $A_k^{\mathrm{answer}}$, where $0 \leq \lambda \leq 1$ is a hyper-parameter controlling the weight of each type of answers, and $S(k)$ is a score function based on the answer rank $k$. If an answer set appears both in executed answer list and generated answer list with ranks $i$ and $j$ respectively, then its score is the sum of two scores: $\lambda S(i) + (1 - \lambda)S(j)$. Finally, we select the answer set with the highest score as the final output. We leave the exploration of other combination methods as future work.

| Model | Overall | I.I.D. | Compositional | Zero-Shot |
|---|---|---|---|---|
| QGG (Lan & Jiang, 2020) | 36.7 | 40.5 | 33.0 | 36.6 |
| Bert+Ranking (Gu et al., 2021) | 58.0 | 67.0 | 53.9 | 55.7 |
| ReTrack (Chen et al., 2021a) | 65.3 | 87.5 | 70.9 | 52.5 |
| S2QL (Zan et al., 2022) | 66.2 | 72.9 | 64.7 | 63.6 |
| ArcaneQA (Gu & Su, 2022) | 73.7 | - | 75.3 | 66.0 |
| RnG-KBQA (Ye et al., 2022) | 74.4 (76.9) | 89.0 (88.3) | 71.2 (69.2) | 69.2 (75.1) |
| RnG-KBQA (T5-large)* | - (77.1) | - (88.5) | - (69.8) | - (75.2) |
| UniParser (Liu et al., 2022) | 74.6 | - | 71.1 | 69.8 |
| DeCC (Anonymous) | 77.6 | - | 75.8 | 72.5 |
| TIARA (Shu et al., 2022) | 78.5 | - | 76.5 | **73.9** |
| DECAF (BM25 + FiD-large) | 76.0 (78.7) | **90.5** (90.2) | 79.0 (78.7) | 68.0 (73.7) |
| DECAF (DPR + FiD-large) | - (75.4) | - (89.7) | - (75.8) | - (69.0) |
| DECAF (BM25 + FiD-3B) | **78.7** (81.4) | 89.9 (89.7) | **81.8** (80.1) | 72.3 (78.4) |

Table 2: F1 scores on the test split of GrailQA. The numbers in the parentheses are F1 scores on the dev split. * means that we replace the original reader T5-base with T5-large and rerun experiments.

## 4 EXPERIMENT

**Experiment Settings.** We use the full Freebase (Bollacker et al., 2008) data pre-processed by Wang et al. (2021) as the KB for all benchmarks. The total number of entities, relations, and triplets are about 88 million, 20k, and 472 million respectively. The total number of passages after linearization is about 126 million. For the retrieval module of DECAF, we use BM25 implemented by Pyserini (Lin et al., 2021) and Dense Passage Retrieval (DPR) (Karpukhin et al., 2020) with BERT-base Devlin et al. (2019) as question and passage encoders. We train separate DPRs on each dataset following the same training process and use the same model architecture in the original paper. The number of retrieved passages is 100 if not specified. For the reading module, we leverage Fusion-in-Decoder (Izacard & Grave, 2021) based on the T5 (Raffel et al., 2020) model: FiD-large and FiD-3B with 770 million and 3 billion model parameters respectively. For the decoding beam size, we use 10 for FiD-large model and 15 for FiD-3B model.

We evaluate DECAF on four benchmark datasets: WebQSP (Yih et al., 2016), ComplexWebQuestions (CWQ) (Talmor & Berant, 2018), FreebaseQA (Jiang et al., 2019), and GrailQA (Gu et al., 2021). Specifically, GrailQA provides the categories of questions: i.i.d., compositional, and zero-shot, which can be used to evaluate model performance over different levels of generalization. All datasets provide both ground-truth answers and logical forms except FreebaseQA, where our model only generates answers as the final output without using the logical form. More details about these datasets are shown in the appendix. Following previous work (Ye et al., 2022; Das et al., 2021; Oğuz et al., 2022), we evaluate our model based on metrics Hits@1 and F1, where Hits@1 focus on the single top-ranked answer while F1 also considers coverage of all the answers.

### 4.1 MAIN RESULT

We compare with both direct answer prediction and semantic parsing based methods and run 5 independent experiments for FiD-large based DECAF to report mean and standard deviation. We don't do this for DECAF with FiD-3B due to the limitation of computation resource. We denote our model in the form of DECAF ({Retriever} + {Reader}) such as DECAF (BM25 + FiD-large). If the retriever is not specified, it means we choose the better one for each dataset respectively.

As shown in Table 1, DECAF achieves new SOTA on WebQSP and FreebaseQA dataset, outperforming both direct-answer-prediction (first block) and semantic parsing based (second block) baselines. On WebQSP, DECAF improves the previous highest Hits@1 by 3.0% and F1 by 2.3%. Note that one of the best-performing baseline UniK-QA uses STAGG (Yih et al., 2015) for entity linking on WebQSP, which is not publicly available and thus can not be applied to other datasets. Compared to UniK-QA, we see that DECAF (DPR + FiD-large) improves the Hits@1 by 1.6% with the same model size, demonstrating the effectiveness of our method even without entity linking. On FreebaseQA, DECAF improves the SOTA Hits@1 significantly by 15.7%. On CWQ dataset, DECAF

---

[3]We locally set up a Virtuoso triplestore service following the GrailQA paper.

| $\lambda$ | 0.0 | 0.2 | 0.4 | 0.45 | 0.49 | 0.51 | 0.55 | 0.6 | 0.8 | 1.0 |
|---|---|---|---|---|---|---|---|---|---|---|
| $S(k) = 1/k$ | 54.7 | 54.7 | 56.1 | 56.3 | 56.5 | 78.7 | 78.7 | 78.7 | 78.7 | 78.7 |
| $S(k) = B - k + 1$ | 54.7 | 55.6 | 56.4 | 56.5 | 56.5 | 78.3 | 78.3 | 78.4 | 78.6 | 78.7 |

Table 3: F1 scores on GrailQA (dev) using DECAF (BM25 + FiD-large) based on different values of $\lambda$ which is the weight of LF-executed answers in the answer combination function. $S(k)$ is the score function of answer rank $k$, and $B$ is the generation beam size.

| Model | WebQSP | | CWQ | GrailQA (dev) | | | |
|---|---|---|---|---|---|---|---|
| | Hits@1 | F1 | Hits@1 | F1 (O) | F1 (I) | F1 (C) | F1 (Z) |
| DECAF (Answer only) | 74.7 | 49.8 | 50.5 | 54.7 | 59.4 | 38.3 | 59.5 |
| DECAF (LF only) | 74.3 | 74.0 | 55.2 | 72.4 | 88.2 | 76.3 | 63.9 |
| DECAF | 80.7 | 77.1 | 68.1 | 78.7 | 90.2 | 78.7 | 73.7 |
| Non-Executable LF% | 11.3 | | 30.2 | 15.8 | 6.2 | 9.6 | 22.5 |
| DECAF$_{sep}$ (Answer only) | 74.2 | 49.5 | 47.9 | 54.6 | 57.7 | 38.8 | 59.8 |
| DECAF$_{sep}$ (LF only) | 72.7 | 73.1 | 54.3 | 74.0 | 91.4 | 75.2 | 66.0 |
| DECAF$_{sep}$ | 80.6 | 77.1 | 66.5 | 80.3 | 92.9 | 78.9 | 75.3 |
| Non-Executable LF% | 12.3 | | 32.8 | 15.9 | 5.2 | 11.9 | 22.4 |

Table 4: We study the performance of our model when only using generated answers (Answer Only) or executed answers by logical forms (LF Only). O, I, C, Z means overall, i.i.d., compositional and zero-shot. DECAF$_{sep}$ means using a separate reader for answer generation and logical form generation respectively instead of a joint reader. We also show the percentage of questions where none of the generated LFs are executable.

achieves very competitive results, the same as the current SOTA method CBR-KBQA. CBR-KBQA is based on the K-Nearest Neighbors approach, which is complementary to our method. We also see that increasing reader size from large (770M) to 3B significantly improves model performance. Moreover, BM25 and DPR lead to different results: DPR performs significantly better than BM25 on WebQSP, slightly better on FreebaseQA, and worse on CWQ. The possible reason is that DPR is trained to retrieve passages containing the answers based on distant supervision (Karpukhin et al., 2020), which do not necessarily contain the relations and entities that appeared in the logical form, especially for complex questions. Thus it may hurt the performance of logical form generation which results in a final performance degeneration.

Table 2 shows results on the GrailQA dataset, where we listed F1 scores of Overall, I.I.D., Compositional, and Zero-shot questions respectively. We see that with FiD-large reader, DECAF achieves better overall performance than the published SOTA method RnG-KBQA. Since original RnG-KBQA uses T5-base, we also equip it with T5-large for a fair comparison and list the results on the dev set, where DECAF (BM25 + FiD-large) still significantly outperforms it by 1.6% in overall F1 score. With FiD-3B reader, the overall F1 of DECAF is improved by 2.7% compared to using FiD-large reader, and surpasses the current SOTA method TIARA[4], which is an anonymous submission on the GrailQA leaderboard. Notably, DECAF performs the best in compositional questions with an F1 score 5.3% higher than the best-performing method.

Overall we see that DECAF achieves new SOTA results on WebQSP, FreebaseQA, and GrailQA with very competitive results on the CWQ benchmark. Even with simple BM25 retrieval, DECAF outperforms most of the baselines across all the benchmark datasets, which previous studies usually use different entity linking methods specially designed for different datasets.

## 4.2 ABLATION STUDY

In this section, we conduct some ablation studies to answer the following questions:

**What is the best way to combine LF-executed answers and generated answers?** In Section 3.4, we introduced a way of combining LF-executed answers and generating answers to obtain the final answer. (1) When none of the LFs are executable, we use the top-1 generated answer as output. In Table 4, we show the percentage of questions where none of the generated LFs are executable.

---

[4]We achieve the 1st rank on the GrailQA leaderboard as of 09/06/2022.

| Model | Combination Type | WebQSP | GrailQA (dev) |
|---|---|---|---|
| ArcaneQA | None | 75.6 | 76.9 |
| ArcaneQA + DECAF (FiD-large, Answer Only) | LF + Answer | 75.8 | 77.4 |
| ArcaneQA + DECAF (FiD-3B, Answer Only) | LF + Answer | 75.8 | 77.5 |
| ArcaneQA + RnG-KBQA | LF + LF | 76.0 | 77.0 |
| RnG-KBQA | None | 76.2 | 77.1 |
| RnG-KBQA + DECAF (FiD-large, Answer Only) | LF + Answer | 76.9 | 77.1 |
| RnG-KBQA + DECAF (FiD-3B, Answer Only) | LF + Answer | 77.0 | 77.1 |
| RnG-KBQA + ArcaneQA | LF + LF | 77.2 | 77.1 |
| DECAF (FiD-large) | LF + Answer | 77.1 | 78.7 |
| DECAF (FiD-3B) | LF + Answer | 78.8 | 81.4 |

Table 5: Results of the combination of baseline methods and comparison with our proposed model DECAF. X + Y means that we first use the answer produced by X as the output; if no answer is produced, we use the answer produced by Y. F1 scores are reported for both WebQSP and GrailQA (dev) datasets.

It can be observed that directly generating LFs for hard questions, such as CWQ and Zero-shot GrailQA, shows a significantly higher non-executable rate than that for easy questions. (2) If any LF is executable, we use the answer with the highest combination scores $\lambda S(i) + (1 - \lambda)S(j)$, where $\lambda$ is the hyper-parameter weight for LF-executed answers, and $i$ and $j$ are the rank in the LF-executed answer list and generated answer list respectively. We test two different score functions $S(k) = 1/k$ and $S(k) = B - k + 1$ where $B$ is the beam size, and $k$ is the rank in the candidate list. From the results in Table 3, we see that the model performance is improved when $\lambda$ increases. Specifically (1) the F1 score increases dramatically when $\lambda$ changes from 0.49 to 0.51, where the former usually chooses generated answer as a final answer while the latter selects the LF-executed one. This demonstrates that LF-executed answers are much more accurate compared to generated ones, which can also be validated in the next section. (2) $\lambda = 1.0$ gives the best results, which means that **we can simply select the first executed answer if exists, otherwise choose the first generated answer**. Results on WebQSP and CWQ show the same patterns, as listed in Appendix A.5. Thus we set $\lambda = 1.0$ as the default setting of DECAF in all the experiments.

**How do answer generation and LF generation perform respectively?** DECAF combines generated answers and executed answers to obtain the final answers. In this section, we study the impact of them separately. We introduce two new models to (1) only use generated answers (Answer Only) and (2) only use executable answers by logical form (LF Only). As shown in the first group of models in Table 4, the performance of LF Only is consistently better than Answer Only. Except on WebQSP, the latter has similar Hits@1, but the former has a significantly better F1 score, which shows that LF can produce much better answer coverage. We also see that the combination of these two strategies is significantly better than any of them separately. For example, on the dev set of GrailQA, the overall F1 of DECAF is 6.3% higher than DECAF (LF only) and 24.0% higher than DECAF (Answer Only). In Appendix A.4, we provide a detailed comparison between the results obtained from logical-form and answer generation methods. This comparison covers various question types, number of relations present in the logical form, and the total number of correct answers.

**Should we use a joint reader or separate readers for answer and LF generation?** We add another variation of our model called $DECAF_{sep}$, which uses two separate readers to generate direct answers and logical forms respectively instead of a shared reader, while other parts remain the same as DECAF. As shown in the second group of models in Table 4, we see that the overall performance is similar to DECAF, showing that a shared reader with prefix is as capable as two separate readers. However, it is interesting to see that on CWQ, a shared reader with multi-task training performs better on answer generation and LF generation while on GrailQA, it performs worse on LF generation compared to two separate readers.

**How does the ensemble of baseline methods perform?** We select two published best semantic parsing baselines, ArcaneQA (Gu & Su, 2022) and RnG-KBQA (Ye et al., 2022), and ensemble them with our DECAF model. That is, if their generated logical forms are not executable, we use the directly generated answers by DECAF as the final answer. We also ensemble these two baselines: if the logical forms generated by one of them are not executable, the logical forms generated by the other method will be used. As shown in Table 5 where F1 scores are reported, we see that they still underperform DECAF. Upon a thorough analysis, we find that, for ArcaneQA and RnG-KBQA,

| Model | WebQSP | | CWQ | | GrailQA | | FreebaseQA | |
|---|---|---|---|---|---|---|---|---|
| | H@100 | R@100 | H@100 | R@100 | H@100 | R@100 | H@100 | R@100 |
| BM25 | 81.2 | 67.7 | 63.5 | 57.7 | 89.9 | 84.7 | 93.5 | 93.5 |
| DPR | 91.6 | 80.6 | 71.4 | 65.6 | 87.6 | 81.0 | 95.0 | 95.0 |

Table 6: Retrieval results (answer name match). H@100 and R@100 stand for the answer hits rate and recall rate of 100 passages, respectively.

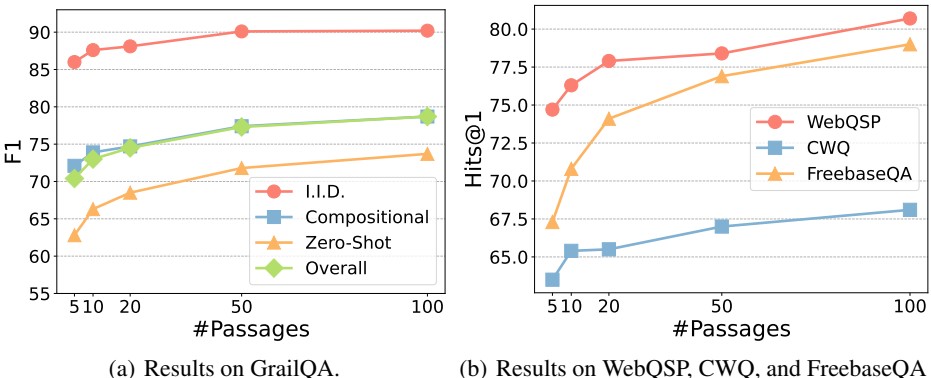

(a) Results on GrailQA.  (b) Results on WebQSP, CWQ, and FreebaseQA

Figure 2: DECAF (FiD-large) performance based on different number of retrieval passages.

very few questions have non-executable logical forms. For ArcaneQA, the corresponding question percentages are 0.7% and 1.4% on WebQSP and GrailQA. For RnG-KBQA, they are 1.8% and 0%. For DECAF (FiD-large), they are 11.3% and 15.8%. The reason is that ArcaneQA uses constrained decoding and RnG-KBQA uses logical form enumeration to increase the execution rate of their generated logical forms. However, a high execution rate does not mean high accuracy. We argue that for some questions, instead of forcing the model to generate executable logical forms, switching to direct-answer generation is a better choice.

**Does retrieval capture useful information for answering questions?** We first evaluate the retrieval results of BM25 and DPR on 4 datasets in Table 6. We observe that: (1) DPR performs better than BM25 on all the datasets except GrailQA, which contains zero-shot questions where DPR may have poor performance due to generalization issues. (2) On WebQSP, GrailQA and FreebaseQA, the retrieval method can achieve over 80% hits rate and recall rate, demonstrating the effectiveness of retrieval. (3) The performance is not as good on CWQ dataset, where most questions require multi-hop reasoning. This problem can be potentially mitigated by iterative retrieval (Xiong et al., 2021; Qi et al., 2021), which we leave as future work. We then analyze the effect of retrieval on the final QA performance in Figure 2, where we show the results over 4 datasets with different numbers of retrieved passages. We see that on all the datasets, with the increase of passage number (from 5 to 100), the model performance is improved. Specifically, on GrailQA dataset, over different categories of questions, we see that the performance over I.I.D. questions increases the least while it improves the most over zero-shot questions. This is because I.I.D. questions can be handled well by memorizing training data while zero-shot questions require KB knowledge to be well answered.

## 5 CONCLUSION

In this paper we propose a novel method DECAF to jointly generate direct answers and logical forms (LF) for knowledge base question answering. We found that combining the generated answers and LF-executed answers can produce more accurate final answers. Instead of relying on entity linking, DECAF is based on a sequence-to-sequence framework enhanced by retrieval, where we transform the knowledge base into text and use sparse or dense retrieval to select relevant information from KB to guide output generation. Experimental results show that we achieve the new state-of-the-art on WebQSP, FreebaseQA, and GrailQA. Our work sheds light on the relationship between more general semantic parsing based methods and direct-answer-prediction methods. It would be interesting to further explore this direction on other tasks involving both semantic parsing and end-to-end methods like table-based question answering or programming code generation.

## 6 ETHICS STATEMENT

We acknowledge the importance of the ICLR Code of Ethics and agree with it. This work retrieves information from linearized KB and decodes logical forms and answers jointly, and uses a combiner to obtain the final answer. One common concern for the KBQA model is that if bias or fairness issues exist in KB, our framework may propagate bias or unfairness when answering the questions based on such KB. In addition, adversarial attacks may alter the behavior and performance of the proposed model in an unexpected way. Therefore, the model should always be used with caution in practice. We believe this work can benefit the field of KBQA, with the potential to benefit other fields involving retrieval-then-reading modeling.

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

## A    APPENDIX

### A.1    KB LINEARIZATION: HYPER-TRIPLET

In Section 3.1, we introduced how we linearize vanilla triplets in KBs. Note that triplets have semantic constrains to express complicated relations. For example, it is hard to express that Richard Nixon and Pat Nixon got married in The Mission Inn Hotel & Spa, which involves three entities. In Freebase, this is solved by introducing a new node called CVT node, which serves as a connecting entity for such hyper-triplets but has no meaning (name) itself. In this example, an entity with id *m.02h98gq* is introduced which involves triplets (*m.02h98gq*, *marriage.spouse*, *Pat Nixon*), (*m.02h98gq*, *marriage.spouse*, *Richard Nixon*), and (*m.02h98gq*, *marriage.location_of_ceremony*, *The Mission Inn Hotel & Spa*). In this case, instead of concatenating CVT node id into sentence, we ignore this node while grouping other entities and relations into one passage: *marriage spouse Richard Nixon. marriage spouse Pat Nixon. marriage location of ceremony The Mission Inn Hotel & Spa.* We illustrate our KB linearization in Figure 3.

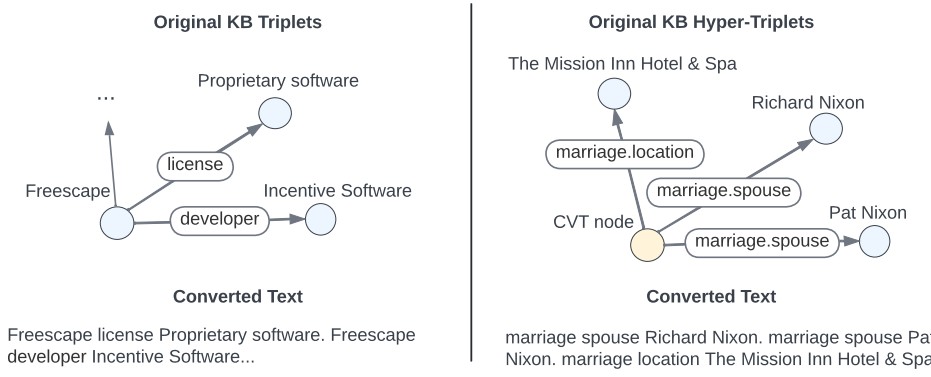

Figure 3: Knowledge base linearization. We show examples of how we linearize triplets (two entities and one relation) and hyper-triplets (multiple entities and relations with a central CVT node).

### A.2    ENTITY NAME DISAMBIGUATION

In KB Linearization (Section 3.1) and logical form generation (Section 3.4), we need to maintain a mapping between an entity ID and an entity name. We can usually assume that one entity ID can be mapped to exactly one entity name, while other names are treated as aliases. However, one entity name can not necessarily be mapped to exactly one entity ID, due to the ambiguity of entity names. For example, in Freebase (Bollacker et al., 2008), the name *Over You* refers to different songs, *Sun* can refer to the star, or an American R&B band. We employ a simple strategy to deal with this ambiguity issue: adding unique suffix *vk* where $k$ is the number of current entities that has the same name. For example if there are three entities named as *Sun*, we'll rename them as *Sun*, *Sun v1*, *Sun v2*. The order of renaming does not matter as long as the names are already differentiated. In this case, during knowledge base linearization, instead of concatenating the original name of entities into sentences, we concatenate their disambiguated names. During model generation, the output entity name can be easily mapped to the corresponding entity ID.

### A.3    MULTI-ANSWER GENERATION

One real-world question can involve multiple correct answers. Like *What are the countries with over 1 billion population?*, the correct answer includes both *China* and *India*. This is also true in the benchmark datasets, as shown in Table 7. We see that only about 50% questions in WebQSP have a single answer, and about 12% contains even over 10 correct answers. It's challenging for direct-answer-generation methods to generate multiple answers accurately. Although DECAF generate a single answer for the direct answer generation, we also explore the possibility of multi-answer generation here. First we tried returning top-$K$ ($K > 1$) generated answers for each question with

beam search. However since the numbers of correct answers to different questions are different, we find that increasing $K$ even degrades the model performance. Then we tried to generate the concatenation of multiple answers splitted by a special token "|". For example, for the question *What are the countries with over 1 billion population?*, the generated output will be "China | India". After generation we split the output by the special token to obtain all the answers. We denote the original (default) method as DECAF (Single Answer) while this one as DECAF (Multiple Answers). The results are shown in Table 8, where we see that generating multiple answers can improve the Answer Only performance, but not the overall performance since it hurts the performance of logical form generation. Thus in our final model, we still use single-answer generation. We leave the exploration of multi-answer generation as future work.

| Dataset | $a = 1$ | $2 \leq a \leq 4$ | $5 \leq a \leq 9$ | $a \geq 10$ |
|---|---|---|---|---|
| WebQSP | 51.2% | 27.4% | 8.3% | 12.1% |
| CWQ | 70.6% | 19.4% | 6.0% | 4.0% |
| GrailQA | 68.4% | 16.2% | 5.3% | 9.9% |
| FreebaseQA | 90.7% | 9.0% | 0.3% | 0% |

Table 7: Percentages of questions containing $a$ answers in each dataset.

| Model | GrailQA (dev) | | | |
|---|---|---|---|---|
| | F1 (O) | F1 (I) | F1 (C) | F1 (Z) |
| DECAF (Single Answer) | 78.7 | 90.2 | 78.7 | 73.7 |
| - Answer Only | 54.7 | 59.4 | 38.3 | 59.5 |
| - LF only | 72.4 | 88.2 | 76.3 | 63.9 |
| DECAF (Multiple Answers) | 78.1 | 93.5 | 76.7 | 71.9 |
| - Answer Only | 60.9 | 70.7 | 44.4 | 63.6 |
| - LF Only | 69.4 | 91.9 | 73.7 | 57.8 |

Table 8: Results comparison between single-answer generation and multi-answer generation in the direct-answer-generation part of DECAF (BM25 + FiD-large).

### A.4 DIRECT COMPARISON BETWEEN LF-EXECUTED AND GENERATED ANSWERS

Based on DECAF (FiD-large) over the GrailQA (dev) dataset, we directly compare the LF-executed answers and direct-generated answers on different question categories. Table 9 shows 4 different comparison results, where we calculate the percentages of questions that they perform equally well, LF better, Answer better and they both have the F1 score of 0. We see that direct-answer generation is more advantageous on the zero-shot questions (15.3%) compared to I.I.D (4.4%). and compositional (6.7%) ones. This also corresponds to the second example in Table 14 where we see that, for zero-shot questions with schemas unseen during training, generating the correct logical forms containing the schemas can be very difficult.

We also conduct a similar analysis based on the number of relations in the ground-truth logical form of each question. Table 10 shows that direct-answer generation has more advantages when the number of relations increases (7.3% → 18.8% → 29.1%), where the logical forms become more complicated and harder to generate.

Finally, we conduct an analysis based on the number of ground-truth answers of each question. In Table 11, we see that logical-form generation has more advantages when the number of answers increases (from 14.8% to 82.7%). This is reasonable since the difficulty of logical form generation is not necessarily correlated with the number of answers, while this is not the case for direct-answer generation.

### A.5 ADDITIONAL ABLATION STUDIES

**What is the best way to combine LF-executed answers and generated answers?** Following Section 4.2, we report the results of answer combination over WebQSP and CWQ datasets. As shown in Table 12, increasing $\lambda$ can improve model performance. The optimal value is $\lambda = 1$ which

| Question Category | Overall | I.I.D. | Compositional | Zero-Shot |
|---|---|---|---|---|
| LF better | 33.0% | 40.6% | 48.4% | 23.4% |
| Answer better | 10.8% | 4.4% | 6.7% | 15.3% |
| Equal F1 | 39.4% | 47.2% | 28.1% | 40.8% |
| Zero F1 | 16.7% | 7.8% | 16.8% | 20.5% |

Table 9: We compare the F1 score per question of LF-executed answers and generated answer based on the category of each question.

| #Relation | 1 | 2 | $\geq 3$ |
|---|---|---|---|
| LF better | 33.1% | 35.7% | 17.9% |
| Answer better | 7.3% | 18.8% | 29.1% |
| Equal F1 | 46.4% | 22.6% | 9.1% |
| Zero F1 | 13.3% | 22.8% | 43.9% |

Table 10: We compare the F1 score per question of LF-executed answers and generated answer based on the number of relations in the ground-truth logical form of each question.

| #Answer | 1 | 2-4 | 5-9 | $\geq 10$ |
|---|---|---|---|---|
| LF better | 14.8% | 71.4% | 74.4% | 82.7% |
| Answer better | 10.6% | 13.0% | 12.2% | 8.1% |
| Equal F1 | 56.2% | 1.0% | 0.3% | 0% |
| Zero F1 | 18.4% | 14.6% | 13.1% | 9.2% |

Table 11: We compare the F1 score per question of LF-executed answers and generated answer based on the number of ground-truth answers of each question.

means that we can simply choose the top-1 LF-executed answer as the final answer if any of the logical forms is executable. If none of them is executable, we still use the directly-generated answer.

| WebQSP / $\lambda$ | 0.0 | 0.2 | 0.4 | 0.45 | 0.49 | 0.51 | 0.55 | 0.6 | 0.8 | 1.0 |
|---|---|---|---|---|---|---|---|---|---|---|
| $S(k) = 1/k$ | 49.8 | 49.8 | 50.6 | 51.0 | 51.3 | 76.7 | 76.7 | 76.9 | 77.1 | 77.1 |
| $S(k) = B - k + 1$ | 49.8 | 51.0 | 51.5 | 51.8 | 51.9 | 75.2 | 75.2 | 75.2 | 76.0 | 77.1 |
| CWQ / $\lambda$ | 0.0 | 0.2 | 0.4 | 0.45 | 0.49 | 0.51 | 0.55 | 0.6 | 0.8 | 1.0 |
| $S(k) = 1/k$ | 50.5 | 50.5 | 52.7 | 53.5 | 54.4 | 68.5 | 68.5 | 68.6 | 68.6 | 68.6 |
| $S(k) = B - k + 1$ | 50.5 | 53.1 | 54.3 | 54.6 | 54.6 | 68.3 | 68.3 | 68.4 | 68.6 | 68.6 |

Table 12: F1 scores on WebQSP and Hits@1 scores on CWQ using DECAF (FiD-large) based on different values of $\lambda$, which is the weight of LF-executed answers in the answer combination function. $S(k)$ is the score function of answer rank $k$ and $B$ is the generation beam size.

**Oracle combination between LF-executed answers and generated answers.** We conduct the oracle combination by selecting the better answer from the direct generation and LF-execution based on its F1 score with the ground-truth answer. In Table 13, we report the oracle results compared to the original results. We see that the oracle combination has better performance compared to our original model, but not that much ($\leq 3$ points improvement), indicating that our current combination method, first LF execution then direct generation, is a simple but very effective choice.

**How does the size of training data affect the model performance?** We study the influence of the size of the training data. Specifically, we want to study how it influences answer generation and LF generation respectively. We focus on the GrailQA dataset, and vary the number of training data from 500, 2000, 10000 to 44337 (all). As shown in Figure 4(a), the performance of DECAF improves as the increase of training data. More importantly, we see that the performance of LF generation improves much more significantly than DECAF (Answer Only). This shows that training the logical form generation requires more data than answer generation, because logical forms are usually more complicated than answers.

**During inference, what is the effect of beam size?** In this section, we study the effects of generation beam size during inference. As shown in Figure 4(b), when we vary the beam size from 1,2,5,10

| Model | WebQSP | CWQ | GrailQA (dev) | | | |
|---|---|---|---|---|---|---|
| | F1 | Hits@1 | F1 (O) | F1 (I) | F1 (C) | F1 (Z) |
| DECAF (FiD-large) | 77.1 | 68.1 | 78.7 | 90.2 | 78.7 | 73.7 |
| DECAF (FiD-large, Oracle) | 79.6 | 70.1 | 80.7 | 91.4 | 80.5 | 76.1 |
| DECAF (FiD-3B) | 78.8 | 70.4 | 81.4 | 89.7 | 80.1 | 78.4 |
| DECAF (FiD-3B, Oracle) | 81.8 | 72.1 | 83.6 | 91.3 | 81.6 | 81.0 |

Table 13: Comparison between our model with original combination method (first LF-executed answers then direct-generated answers) and our model with oracle combination method where we select the answer with better F1 according to the ground-truth answers.

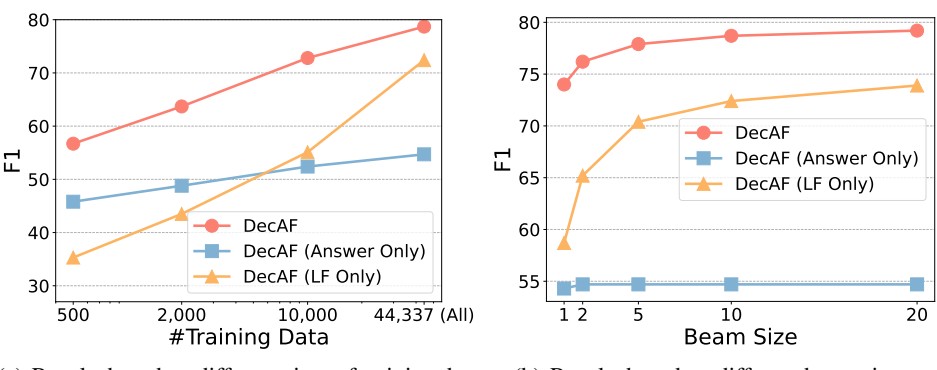

(a) Results based on different sizes of training data    (b) Results based on different beam sizes

Figure 4: Ablation study on training data size and generation beam size over GrailQA (dev) dataset.

to 20, the performance of DECAF improves. However, we see that the performance DECAF (LF Only) is significantly improved while DECAF (Answer Only) barely changes. This shows that the overall performance improvement mainly comes from logical form generation instead of answer generation. This is because the logical form is difficult to generate, and beam size is important to long sequence generation especially when we enumerate the generated logical forms until we find the one that is executable. However, for answer generation, which is usually in short length, beam size won't have a large effect. We also show the results of different beam sizes on WebQSP and CWQ datasets in Figure 5, where we have similar observations to GrailQA.

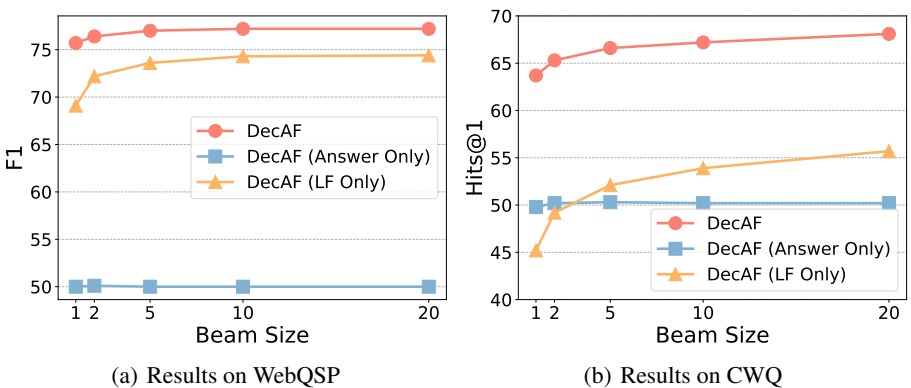

(a) Results on WebQSP    (b) Results on CWQ

Figure 5: Ablation study on generation beam size using DECAF (FiD-large).

## A.6 ERROR ANALYSIS

We conduct case-based error analysis on our model. As shown in Table 14, we list 3 cases where Answer Only generation is wrong, LF Only generation is wrong, and both of them are wrong.

In the first example, we need to find the "earlies" composition. We see that the logical form is not complicated while the direct answer generation needs to reason over the completion time of compositions and find the minimum of them, which is difficult. In the second example, we see that the reasoning is not difficult since it only contains one relation about locomotive class, but the logical form is relatively long which makes the generation more challenging. We see that the generated logical form misses the *steam* part and results in an non-executable prediction. For these two cases, DECAF can still output the correct answer due to answer combination. However, this is not the case for the last example, which is a compositional question involving multiple relations. We see that both the generated answer and logic form are wrong. The generated logical form neglects one join operation and makes mistake on the relation about release date. This means that our model can be further improved to deal with such complicated questions.

| | |
|---|---|
| Question: | *Which composition was completed the earliest?* |
| Gold Answer: | *Ce fut en mai* |
| Gold Logical Form: | `(ARGMIN music.composition music.composition.date_completed)` |
| Gen Answer: | *Composition for Piano and Orchestra by David Bowie* |
| Gen Logical Form: | `(ARGMIN music.composition music.composition.date_completed)` |
| LF Executed Answer: | *Ce fut en mai* |
| DECAF's Answer: | *Ce fut en mai* |
| Question: | *British rail class 04 belongs to which locomotive class?* |
| Gold Answer: | *0-6-0* |
| Gold Logical Form: | `(AND rail.steam_locomotive_wheel_configuration (JOIN rail.steam_locomotive_wheel_configuration.locomotive_classes m.02rh__)))` |
| Gen Answer: | *0-6-0* |
| Gen Logical Form: | `(AND rail.locomotive_wheel_configuration (JOIN rail.locomotive_wheel_configuration.locomotive_classes m.02rh__)))` |
| LF Executed Answer: | *Not Executable* |
| DECAF's Answer: | *0-6-0* |
| Question: | *Which browser was most recently released by the creators of mpd?* |
| Gold Answer: | *Internet Explorer for Mac* |
| Gold Logical Form: | `(ARGMAX (AND computer.web_browser (JOIN (R computer.software_developer.software) (JOIN (R computer.file_format.format_creator) m.02l0900))) computer.software.first_released)` |
| Gen Answer: | *WebKit* |
| Gen Logical Form: | `(ARGMAX (AND computer.web_browser (JOIN (R computer.software_developer.software) m.02l0900)) computer.software.release_date)` |
| LF Executed Answer: | *Not Executable* |
| DECAF's Answer: | *WebKit* |

Table 14: Case-based error analysis over GrailQA (dev) dataset, where Gen is the abbreviation for Generated. We show 3 cases where only generated answer is wrong, only generated LF is wrong, and both of them are wrong.

