# OpenReview forum: "DecAF: Joint Decoding of Answers and Logical Forms for Question Answering over Knowledge Bases"
_ICLR.cc/2023/Conference — ICLR 2023 poster_

### Official Review · Reviewer_B5YN · 2022-10-25

**Confidence:** 4
**Correctness:** 4
**Technical Novelty And Significance:** 3
**Empirical Novelty And Significance:** 3
**Recommendation:** 8

**Clarity, Quality, Novelty And Reproducibility:**

-

**Strength And Weaknesses:**

-

**Summary Of The Paper:**

This paper proposes DecAF, a model for KBQA. It jointly decodes both logical forms (LF) and text answers and combines the decoded answer(s) with the answer obtained by executing the LF over a KB. The authors show that this results in better performance than other state-of-the-art models on multiple datasets. A brief overview of the method is:
(i) a KB is first linearized into text documents so that text-retrieval methods can be used select relevant subgraphs (BM25/dense retriever is used),
(ii) a T5-based FiD is used to encode the concatenation of the question and passages (multiple passages are retrieved, so this is performed for each (q,p_i) pair),
(iii) this model is trained to generate both LF and text  answers by prompting it with suitable prefixes,
(iv) a simple scoring function is used to choose the best answer from a list of candidates containing both direct and LF-executed answers.

There are quite a few advantages of this method:
(i) no explicit entity linking is required,
(ii) direct decoding of answers is used as a fallback when valid LFs are not generated, thereby increasing the chances of getting the right answer,
(iii) choosing relevant subgraphs has been a difficult problem to solve, and treating it as a text-retrieval problem sidesteps this,
(iv) method is general enough to be applied to other tasks like QA on tables, etc.

Overall the results are promising, though it struggles a bit in the zero-shot setting. The analysis and ablations answer most if not all questions which naturally arise from the Method section.
Question/concern regarding the linearization of the KG: LLMs (in this case T5) are pre-trained on well-formed sentences. However, the retrieved subgraph is the form of <head relation tail> which is not well-formed. How will this affect its representation?

As such I don't find any other weaknesses in the paper.

**Summary Of The Review:**

-

---

> ### Author Response · Authors · 2022-11-16
> **Response to Reviewer B5YN**
>
> Thanks for your positive comments and we'll keep improving our paper! Our response has been provided below.
>
> > LLMs (in this case T5) are pre-trained on well-formed sentences. However, the retrieved subgraph is the form of <head relation tail> which is not well-formed. How will this affect its representation?
>
> In our experiment, we found that it can capture the information of the retrieved subgraphs very well. As shown in Figure 2 on Page 9, increasing the number of retrieved passages can significantly improve the model performance, which demonstrates that T5 is effectively leveraging the information from those passages. We think there are two reasons: 1. T5 was pre-trained on a large amount of web data, which includes both structured and unstructured data (in some cases even noisy data). 2. In our work, we further fine-tune T5 on KBQA datasets, which allows it to further adapt to our retrieved subgraph during training. We think both factors are important and It would be interesting to study this in the zero-shot or few-shot settings where very little downstream data is available.

---

### Official Review · Reviewer_9YKE · 2022-10-26

**Confidence:** 3
**Correctness:** 4
**Technical Novelty And Significance:** 3
**Empirical Novelty And Significance:** 4
**Recommendation:** 5

**Clarity, Quality, Novelty And Reproducibility:**

The writing and presentation of this paper are good, and the proposed method is well-motivated. Mitigating the non-execution issue of logical form generation is a problem of significance in semantic parsing-based KBQA methods. The incorporation of a retriever and Fushion-in-Decoder is a good design. But the way to combine executing results of logical forms and answers directly generated is straightforward, lacking technical novelty to some extent. Performance improvement over logical form-based methods can be easily expected by simply using a weighted ranking score, especially when logical forms are not executable. Related to this concern, how these two types of generation targets complement each other deserves more in-depth discussion, in addition to the analyses on lambda. For example,  in addition to considering the executability of logical forms, on what kinds of questions can each method demonstrate its unique merits？ And how does the complementary effect form?


**Strength And Weaknesses:**

**Strengths**
* The motivation for combining the merits of logical forms and direct answers is reasonable.
* The method demonstrates strong performance, especially on WebQSP, FreebaseQA, and GrailQA benchmarks.
* The method avoids the entity linking step to locate entities by linearizing knowledge bases and incorporating retrieval.
* The paper is well-structured and easy to follow.

**Weaknesses**
* The combination mechanism of logical forms and direct answers lacks novelty.
* The collaboration between logical forms and direct answers can be analyzed more in-depth.

**Summary Of The Paper:**

This paper presents a method named DECAF for Question answering over knowledge bases. DECAF first retrieves relevant facts in KBs and then leverages multiple prompts to generate logical forms and direct answers using a Fushion-in-Decoder architecture. The final answer entity is produced by combining these two types of outputs. Experiments on benchmarks of WebQSP, FreebaseQA, GrailQA, and ComplexWebQuestions demonstrate the effectiveness of the proposed method.


**Summary Of The Review:**

This paper proposes a pipelined method for question answering over KBs by jointly generating logical forms and direct answers. The motivation is clear, and the technique is empirically effective. My main concern is the lack of novelty in the whole pipeline in terms of integrating direct answers and the ones generated by logical forms. Analyses of the integration effects are also somewhat weak.

---

> ### Author Response · Authors · 2022-11-16
> **Response to Reviewer 9YKE**
>
> Thanks for your questions and suggestions. We have provided our responses below.
>
> > 1. The combination mechanism of logical forms and direct answers lacks novelty.
>
> We want to emphasize the novel contributions of our paper:
> 1. We are the first to propose a unified KBQA framework that combines the merits of both semantic parsing and direct question answering.
> 2. We remove entity linking, which is commonly used by previous methods, and use text-based retrieval to enhance the adaptability of our model to new domains.
> 3. Our method achieves new state-of-the-arts on 3 benchmark datasets, with significant improvements in F1 scores over WebQSP (2.3%) and FreebaseQA (15.7%), and 1st rank on the GrailQA leaderboard.
>
> > 2. The collaboration between logical forms and direct answers can be analyzed more in-depth.
>
> We’ve added additional analysis to our paper in Appendix A.4, A.5, and A.6. Please kindly refer to our general response and revised paper.
>
> > 3. In addition to considering the executability of logical forms, on what kinds of questions can each method demonstrate its unique merits? And how does the complementary effect form?
>
> We directly compare the LF-executed answers and direct-generated answers on different question categories of the GrailQA (dev) dataset. The table below shows 4 different comparison results, where we calculate the percentages of questions that they perform equally well, LF better, Answer better and they both have the F1 score of 0. We see that direct-answer generation is more advantageous on the zero-shot questions (15.3%) compared to I.I.D (4.4%) and compositional (6.7%) ones. This also corresponds to the second example in Table 14 where we see that for zero-shot questions with schemas unseen during training, generating the correct logical forms containing the schemas can be very difficult.
>
> |               | Overall |  I.I.D. | Compositional | Zero-Shot |
> |---------------|---------|---------|---------------|-----------|
> | LF better     | 33.0%   | 40.6%   | 48.4%         | 23.4%     |
> | Answer better | 10.8%   | 4.4%    | 6.7%          | 15.3%     |
> | Equal F1      | 39.4%   | 47.2%   | 28.1%         | 40.8%     |
> | Zero F1       | 16.7%   | 7.8%    | 16.8%         | 20.5%     |
>
> We also conduct a similar analysis based on the number of relations in the ground-truth logical form of each question. The table below shows that direct-answer generation has more advantages when the number of relations increases (7.3% -> 18.8% -> 29.1%), where the logical forms become more complicated and harder to generate.
>
> | # Relation    | 1     |  2    | >= 3  |
> |---------------|-------|-------|-------|
> | # Question    | 4950  | 1528  | 285   |
> | LF better     | 33.1% | 35.7% | 17.9% |
> | Answer better | 7.3%  | 18.8% | 29.1% |
> | Equal F1      | 46.4% | 22.6% | 9.1%  |
> | Zero F1       | 13.3% | 22.8% | 43.9% |
>
> We further conduct analysis based on the number of ground-truth answers to each question. The table below shows that logical-form generation is more advantageous when the number of answers increases (from 14.8% to 82.7%). This is reasonable since the difficulty of logical form generation is not necessarily correlated with the number of answers while this is not the case for direct-answer generation.
>
> | # Answer      | 1     |  2-4  | 5-9   | 10+   |
> |---------------|-------|-------|-------|-------|
> | # Question    | 4718  | 1114  | 352   | 579   |
> | LF better     | 14.8% | 71.4% | 74.4% | 82.7% |
> | Answer better | 10.6% | 13.0% | 12.2% | 8.1%  |
> | Equal F1      | 56.2% | 1.0%  | 0.3%  | 0%    |
> | Zero F1       | 18.4% | 14.6% | 13.1% | 9.2%  |

---

> ### Author Response · Authors · 2022-12-01
> **Response to Reviewer 9YKE (Follow-up)**
>
> We greatly appreciate your effort and time in reviewing our work and providing constructive feedback!
> We are following up to check whether our responses have addressed your comments/concerns.
> Thank you, Authors

---

### Official Review · Reviewer_Azh2 · 2022-10-28

**Confidence:** 4
**Correctness:** 3
**Technical Novelty And Significance:** 2
**Empirical Novelty And Significance:** 2
**Recommendation:** 6

**Clarity, Quality, Novelty And Reproducibility:**

Clarity: The paper is well-written and was easy to follow

Quality: The proposed method is simple and the efficacy of the model is tested on multiple KBQA benchmarks. I would say that the work is of high quality

Novelty: The technical novelty of the work is rather limited.

Reproducibility: The authors promise that the code will be opensourced. The retriever and reader models are fairly standard and well-known and should aid in reproducibility

**Strength And Weaknesses:**

**Strengths**

- There is a growing body of work in QA literature which are unifying the models that work for different QA modalities - text, KB, tables. This paper is another example that shows powerful models developed for textual QA are effective for KBQA.
- The paper presents a simple and straightforward method that is effective on multiple KBQA benchmarks. Also developing good entity linking models has been challenging needing a lot of supervised data. Aiming to solve it by textual retrieval is simple and interesting.
- The paper was easy to follow.

**Weaknesses**

- A question that this paper does not address is the intuition behind why training a model to produce both logical forms and direct answers is expected to work better? Is it because for some questions it is easier to predict the logical forms and for some it is easier to directly decode the answer and hence it is best to combine both? I think the paper would benefit with an intuition behind why the model is working better
- The technical novelty of the paper is limited. Even though the proposed method is simple (which I believe is a strength), the use of textual QA models is not novel. The training for producing both logical forms and direct answers is new, however is presented without much motivation.
- Question: These datasets contain many questions which have multiple answers. How do you handle that during training and testing? Is the direct-decode model trained to produce all answers?

**Summary Of The Paper:**

This paper proposes a retriever+reader model for knowledge-base question answering (KBQA) where the retriever and readers are models that are usually used for open-domain textual question answering - e.g. BM25/DPR models for retriever and FiD model as the reader. The KB triples are first linearized into text. Each triple (e1, r, e2) is converted into the textual format by using concatenating the textual name of entities and using a space delimiter (e.g. (Freescape, game engine.developer, Incentive Software) → Freescape game engine developer Incentive Software). Next, triples for an entity are grouped into a single document. Following DPR, each document is split into multiple paragraphs (each 100 tokens)

Given a query, a retriever first retrieves paragraphs (which are essentially triples). This is followed by feeding the retrieved paragraphs to an encoder-decoder FiD reader model. The technical contribution of the paper is to train a model to produce both the logical forms and directly the answers. Depending on the output type, a separate prompt is prepended to the input.

The use of the textual form for entity names also eliminates the requirement of an entity linker model which links the mentions of entities in the question to entities in the KB. Instead, that is handled by the retriever which uses lexical matching to get the relevant triples.

The answers obtained by executing the generated logical form and the answers that are generated directly are further combined using a simple weighted addition.

This proposed simple method does well on multiple KBQA benchmarks - WebQSP, CWQ, GrailQA, FreebaseQA.

**Summary Of The Review:**

I think this is a simple model that works well. I think the paper will improve if it addresses why training to generate both the logical forms and answers would be better. Moreover, apart from this, the techinical novelty of the paper is rather limited. Hence, I am leaning towards a weak accept score.

---

> ### Author Response · Authors · 2022-11-16
> **Response to Reviewer Azh2**
>
> Thanks for your questions and suggestions. We have provided our responses below.
>
> > 1. intuition behind why training a model to produce both logical forms and direct answers is expected to work better? Is it because for some questions it is easier to predict the logical forms and for some it is easier to directly decode the answer and hence it is best to combine both?
>
> Correct. We conduct more in-depth analysis below.
>
> We directly compare the LF-executed answers and direct-generated answers on different question categories of the GrailQA (dev) dataset. The table below shows 4 different comparison results, where we calculate the percentages of questions that they perform equally well, LF better, Answer better and they both have the F1 score of 0. We see that direct-answer generation is more advantageous on the zero-shot questions (15.3%) compared to I.I.D (4.4%) and compositional (6.7%) ones. This also corresponds to the second example in Table 14 where we see that for zero-shot questions with schemas unseen during training, generating the correct logical forms containing the schemas can be very difficult.
>
> |               | Overall |  I.I.D. | Compositional | Zero-Shot |
> |---------------|---------|---------|---------------|-----------|
> | LF better     | 33.0%   | 40.6%   | 48.4%         | 23.4%     |
> | Answer better | 10.8%   | 4.4%    | 6.7%          | 15.3%     |
> | Equal F1      | 39.4%   | 47.2%   | 28.1%         | 40.8%     |
> | Zero F1       | 16.7%   | 7.8%    | 16.8%         | 20.5%     |
>
> We also conduct a similar analysis based on the number of relations in the ground-truth logical form of each question. The table below shows that direct-answer generation has more advantages when the number of relations increases (7.3% -> 18.8% -> 29.1%), where the logical forms become more complicated and harder to generate.
>
> | # Relation    | 1     |  2    | >= 3  |
> |---------------|-------|-------|-------|
> | # Question    | 4950  | 1528  | 285   |
> | LF better     | 33.1% | 35.7% | 17.9% |
> | Answer better | 7.3%  | 18.8% | 29.1% |
> | Equal F1      | 46.4% | 22.6% | 9.1%  |
> | Zero F1       | 13.3% | 22.8% | 43.9% |
>
> We further conduct analysis based on the number of ground-truth answers to each question. The table below shows that logical-form generation is more advantageous when the number of answers increases (from 14.8% to 82.7%). This is reasonable since the difficulty of logical form generation is not necessarily correlated with the number of answers while this is not the case for direct-answer generation.
>
> | # Answer      | 1     |  2-4  | 5-9   | 10+   |
> |---------------|-------|-------|-------|-------|
> | # Question    | 4718  | 1114  | 352   | 579   |
> | LF better     | 14.8% | 71.4% | 74.4% | 82.7% |
> | Answer better | 10.6% | 13.0% | 12.2% | 8.1%  |
> | Equal F1      | 56.2% | 1.0%  | 0.3%  | 0%    |
> | Zero F1       | 18.4% | 14.6% | 13.1% | 9.2%  |
>
> > 2. These datasets contain many questions which have multiple answers. How do you handle that during training and testing? Is the direct-decode model trained to produce all answers?
>
> The current direct-decode model only generates a single answer instead of multiple answers together. During training, we randomly sample an answer as the training target for each training step. During testing, we choose the top-1 generated answer using beam search. We also tried to generate multiple answers as discussed in Appendix A.3, where we find that it can improve the performance of direct-answer generation, but the final performance after combination with logical-form generation is not improved. We think the reason is that the performance improvement of multi-answer generation comes from questions where logical forms can also perform very well. We leave the exploration of more advanced multi-answer generation methods as future work.

---

### Official Review · Reviewer_ZmkY · 2022-10-29

**Confidence:** 3
**Correctness:** 3
**Technical Novelty And Significance:** 2
**Empirical Novelty And Significance:** 3
**Recommendation:** 6

**Clarity, Quality, Novelty And Reproducibility:**

The paper is well-written. Methods, experimental setup, and analyses are described clearly and should be reproducible.



**Strength And Weaknesses:**

Strengths:
1. The main idea of combining two types of answering methods is simple, well-motivated and turns out to be effective.

2. Having a single reader that can do both logical form generation and direct answer generation, rather than having two separate readers is efficient.

Weaknesses:

1. While I am convinced that the paper makes a convincing argument for the above contributions, it doesn’t seem interesting enough. It only appears to be a step beyond ensembling, where the systems being ensembled are just two routes to answer from the same reader.

Update after author response: As the authors argue in their rebuttal, this characterization is a bit restrictive and there is value to empirically demonstrating that a single reader can be used to provide two routes.

2. The ablation analyses, unfortunately, don't yield any interesting insights --- what works is to take the answer from the logical form if it executes otherwise use the direct answer; joint training doesn't yield added benefits over separate readers; DPR works better than BM25; more training improves all models; more passages improves; larger beam sizes improve


3. Combining two routes of answering is useful but it doesn't quite shed light on the failure modes of the individual methods. For example, it would be nice to gain insight on the questions for which we can get answers via the direct generation method but not via logical form execution.

Update after author response: The authors have added additional analyses that provide more details on where the combination works and to some extent on why the combination works.

Additional Comments/Questions to the authors:

1. Other types of combinations: Since you are effectively combining two types of methods for generating answers, would it be useful to think about other choices of two methods for combining answers? This could be a simple “ensembling” strategy which serves as a comparison point. The ablation you did in Table 4 is one such instance. For example, you could take one of the baseline methods for direct answer and combine it with answers from one of the execution based methods. You can also combine two methods of the same type also.

2. Oracle experiments: It would be useful to know what the upper-bound performance you can hope for if you allowed an oracle to choose between the top answer from the direct and execution based method? Or is it the case that the execution method if it returns an answer is always the correct one. Examples in Table 9 in Appendix are helpful but it doesn’t quite answer the question above.

3. Question difficulty: It would be useful to know the performance of the direct answer generation accuracy on the set of questions for which the logical form based execution fails. Is this accuracy the same as the accuracy over the entire set of questions? Or is this subset somehow easier/harder for direct answer generation?


**Summary Of The Paper:**

Goal: The paper addresses the brittleness challenge with semantic parsing based QA systems. For some questions models don’t produce bad logical forms that are not executable, which means they get no answer.

Method: Use a direct answer generator. Instead of using a separate answer generator, you can use the same “reader” to generate both the logical form and an answer. If the logical form executes, use its answer. Otherwise, use the direct answer.

Evaluation: The paper compares against models that only use direct answer generation methods or ones that use logical form execution methods, on multiple datasets.

Key Contributions:
The combination method and an empirical evaluation that shows it works better on multiple datasets.
Ablation studies and analysis that shed some light on joint training for logical form and answer generation, sensitivity to answer combination method, impact of retrieval system, and beam search choices.


**Summary Of The Review:**

The main idea is well-motivated. The basic idea that combining directly generated answer when logical form based execution fails is useful but is expected. The interesting contribution perhaps is in showing we can achieve this without having to ensemble outputs from two separate systems. But otherwise I am struggling to see what new useful information we learn here.

Update after author response: The analyses and experiments from the authors provide additional information that add value to the paper.

---

> ### Author Response · Authors · 2022-11-16
> **Response to Reviewer ZmkY (1/2)**
>
> We appreciate the in-depth questions and suggestions given by the reviewer. We have provided our responses below.
>
> > 1. It only appears to be a step beyond ensembling, where the systems being ensembled are just two routes to answer from the same reader.
>
> We want to emphasize the novel contributions of our paper:
> 1. We are the first to propose a unified KBQA framework that combines the merits of both semantic parsing and direct question answering.
> 2. We remove entity linking, which is commonly used by previous methods, and use text-based retrieval to enhance the adaptability of our model to new domains.
> 3. Our method achieves new state-of-the-arts on 3 benchmark datasets, with significant improvements in F1 scores over WebQSP (2.3%) and FreebaseQA (15.7%), and 1st rank on the GrailQA leaderboard.
>
> We believe our work is not just “a step beyond ensembling”. It’s academically valuable to show that **one** model can generate answers from two different worlds: semantic parsing and direct QA, and the answers are complementary to each other.
>
> > 2. The ablation analyses, unfortunately, don't yield any interesting insights.
>
> We’ve added additional analysis to our paper in Appendix A.4, A.5, and A.6. Please kindly refer to our general response and revised paper.
>
> > 3. it would be nice to gain insight on the questions for which we can get answers via the direct generation method but not via logical form execution.
>
> We directly compare the LF-executed answers and direct-generated answers on different question categories of the GrailQA (dev) dataset. The table below shows 4 different comparison results, where we calculate the percentages of questions that they perform equally well, LF better, Answer better and they both have the F1 score of 0. We see that direct-answer generation is more advantageous on the zero-shot questions (15.3%) compared to I.I.D (4.4%) and compositional (6.7%) ones. This also corresponds to the second example in Table 14 where we see that for zero-shot questions with schemas unseen during training, generating the correct logical forms containing the schemas can be very difficult.
>
> |               | Overall |  I.I.D. | Compositional | Zero-Shot |
> |---------------|---------|---------|---------------|-----------|
> | LF better     | 33.0%   | 40.6%   | 48.4%         | 23.4%     |
> | Answer better | 10.8%   | 4.4%    | 6.7%          | 15.3%     |
> | Equal F1      | 39.4%   | 47.2%   | 28.1%         | 40.8%     |
> | Zero F1       | 16.7%   | 7.8%    | 16.8%         | 20.5%     |
>
> We also conduct a similar analysis based on the number of relations in the ground-truth logical form of each question. The table below shows that direct-answer generation has more advantages when the number of relations increases (7.3% -> 18.8% -> 29.1%), where the logical forms become more complicated and harder to generate.
>
> | # Relation    | 1     |  2    | >= 3  |
> |---------------|-------|-------|-------|
> | # Question    | 4950  | 1528  | 285   |
> | LF better     | 33.1% | 35.7% | 17.9% |
> | Answer better | 7.3%  | 18.8% | 29.1% |
> | Equal F1      | 46.4% | 22.6% | 9.1%  |
> | Zero F1       | 13.3% | 22.8% | 43.9% |

---

> > ### Author Response · Authors · 2022-11-16
> > **Response to Reviewer ZmkY (2/2)**
> >
> > > 4. Other types of combinations: Would it be useful to think about other choices of two methods for combining answers?
> >
> > We select two published best semantic-parsing baseline methods ArcaneQA and RnG-KBQA, and combine them with our direct-answer generation results. That is, if their generated logical forms are not executable, we use the directly generated answers by DecAF as the final answer. We also combine these two baseline methods: if the logical forms generated by one of them are not executable, use the logical forms generated by the other method. As shown in the Table below where F1 scores are reported, we see that they still underperform our DecAF method. After looking at their results in more detail, we find that for ArcaneQA and RnG-KBQA, very few questions have non-executable logical forms. For ArcaneQA, the corresponding question percentages are 0.7% and 1.4% on WebQSP and GrailQA. For RnG-KBQA, they are 1.8% and 0%. For our model DecAF (FiD-large), they are 11.3% and 15.8%. The reason is that ArcaneQA uses constrained decoding and RnG-KBQA uses logical form enumeration to increase the execution rate of their generated logical forms. But a high execution rate does not mean high accuracy. We argue that for some questions (like zero-shot and multi-relational questions as mentioned in Question 1), instead of forcing the model to generate executable logical forms, switching to direct-answer generation is a better choice.
> >
> > | Model                                     | Combination Type | WebQSP | GrailQA (dev) |
> > |-------------------------------------------|------------------|--------|---------------|
> > | ArcaneQA                                  | None             | 75.6   | 76.9          |
> > | ArcaneQA + DecAF (FiD-large, Answer Only) | LF + Answer      | 75.8   | 77.4          |
> > | ArcaneQA + DecAF (FiD-3B, Answer Only)    | LF + Answer      | 75.8   | 77.5          |
> > | ArcaneQA + RnG-KBQA                       | LF + LF          | 76.0   | 77.0          |
> > | RnG-KBQA                                  | None             | 76.2   | 77.1          |
> > | RnG-KBQA + DecAF (FiD-large, Answer Only) | LF + Answer      | 76.9   | 77.1          |
> > | RnG-KBQA + DecAF (FiD-3B, Answer Only)    | LF + Answer      | 77.0   | 77.1          |
> > | RnG-KBQA + ArcaneQA                       | LF + LF          | 77.2   | 77.1          |
> > | DecAF (FiD-large)                         | LF + Answer      | 77.1   | 78.7          |
> > | DecAF (FiD-3B)                            | LF + Answer      | 78.8   | 81.4          |
> >
> > > 5. Oracle experiments: It would be useful to know what upper-bound performance you can hope for if you allowed an oracle to choose between the top answer from the direct and execution based method?
> >
> > We conduct the oracle combination by selecting the better answer from direct generation and LF-execution based on its F1 score with the ground-truth answer. We report the oracle results below compared to the original results. We see that the oracle combination does have better performance compared to our original model, but not that much (<= 3 points improvement), which indicates that our current combination method (first LF execution then direct generation) is a simple but very effective choice.
> >
> > | Model                     | WebQSP | CWQ    | GrailQA (dev) |
> > |---------------------------|--------|--------|---------------|
> > |                           | F1     | Hits@1 | F1            |
> > | DecAF (FiD-large)         | 77.1   | 68.1   | 78.7          |
> > | DecAF (FiD-large, Oracle) | 79.6   | 70.1   | 80.7          |
> > | DecAF (FiD-3B)            | 78.8   | 70.4   | 81.4          |
> > | DecAF (FiD-3B, Oracle)    | 81.8   | 72.1   | 83.6          |
> >
> > > 6. Question difficulty: It would be useful to know the performance of the direct answer generation accuracy on the set of questions for which the logical form based execution fails.
> >
> > We collect, in the table below, the direct answer generation accuracy on the set of questions where the generated logical forms are not executable. From the result, we observe that the performance of direct-answer generation for the data subset for non-executable LF is much lower than the overall evaluation dataset (19.6% F1 drop in WebQSP, 13% Hit@1 drop in CWQ and 14.7% F1 drop in GrailQA). Therefore, it shows that the set of questions for which the logical form based execution fails is also difficult for direct answer generation.
> >
> > | Question Type      | WebQSP | CWQ    | GrailQA (dev) |
> > |--------------------|--------|--------|---------------|
> > |                    | F1     | Hits@1 | F1            |
> > | Overall            | 49.8   | 50.5   | 54.7          |
> > | Non-executable LF  | 30.2   | 37.5   | 40.0          |

---

> > ### Comment · Reviewer_ZmkY · 2022-12-01
> > **Thank you for the follow-up**
> >
> > Thank you for the follow up. I agree the "only a step beyond ensembling" characterization minimizes some of the contributions of the work. Your argument there is a fair one. The analyses you have added provide some insight into what categories of questions for which the different routes work better. The other analyses and results also provide more insights for building on this. I will revise my rating to reflect the updates you have provided. Thank you.

---

> ### Author Response · Authors · 2022-12-01
> **Response to Reviewer ZmkY (Follow-up)**
>
> We greatly appreciate your effort and time in reviewing our work and providing constructive feedback!
> We are following up to check whether our responses have addressed your comments/concerns.
> Thank you, Authors

---

### Author Response · Authors · 2022-11-16
**General Response**

We thank all the reviewers for their precious time and insightful comments. We appreciate that the reviewers recognize our work as well-motivated (Reviewer ZmkY, 9YKE), empirically effective (Reviewer ZmkY, Azh2, 9YKE), and well-written (Reviewer ZmkY, Azh2, 9YKE).

We want to summarize the novel contributions of our work:
1. We are the first to propose a unified KBQA framework that combines the merits of both semantic parsing and direct question answering.
2. We remove entity linking, which is commonly used by previous methods, and use text-based retrieval to enhance the adaptability of our model to new domains.
3. Our method achieves new state-of-the-arts on 3 benchmark datasets, with significant improvements in F1 scores over WebQSP (2.3%) and FreebaseQA (15.7%), and 1st rank on the GrailQA leaderboard.

To improve the paper's quality, we respond to the reviewers’ comments by making the following revisions (marked in blue) to the paper:
1. We add more analysis on what type of questions logical form or answer generation can perform better in Appendix A.5, as suggested by Reviewer ZmkY, Azh2, and 9YKE. We show that compared to semantic parsing, direct-answer generation is preferable when the number of relations in the logical form increases and the number of total answers decreases. We also show that direct-answer generation is more advantageous on the zero-shot questions rather than I.I.D and compositional ones.
2. We add the experiment results of the combination over baselines in Appendix A.4, as suggested by Reviewer ZmkY. We show that our model still outperforms the combination of the strongest semantic parsing baselines and direct-answer generation methods.
3. We add the results of the oracle combination method over LF-executed answers and generated answers in Appendix A.6, as suggested by Reviewer ZmkY. We show that the oracle combination has better performances compared to our original combination, but not that much (<= 3 points improvement), which indicates that our current combination method (first LF execution then direct generation) is a simple but very effective choice.

---

### Decision · Program_Chairs · 2023-01-20

**Decision:**

Accept: poster

**Justification For Why Not Higher Score:**

Limited technical novelty, application might be of limited interest of the ICLR community (in term of number of people).

**Justification For Why Not Lower Score:**

Simple method which works well, solid experimental results, paper well written.

**Metareview: Summary, Strengths And Weaknesses:**

This paper proposes a new method for question answering over knowledge bases (KB), by jointly generating logical forms that can be executed on the KB and by generating the answers directly. The two results are then combined to get the best of both worlds, leading to state-of-the-art results on benchmarks.

Overall, while the reviewers were a bit concerned by the lack of novelty, as the proposed method can be seen as a combination of existing works, they also believed that this method is simple and more efficient than existing work. One interesting contribution of the paper is the removal of entity linking, by instead relying on standard information retrieval over linearised KB. This makes the application of the method easier over new domains or KB, and potentially opens the integration of such systems to more general QA models. Because the method is simple, potentially leading to a more widespread adoption of such techniques, and the empirical results are strong, I recommend to accept the paper.

During the discussion with reviewers, they suggested that some ablations (comparison to ensembling, performance on different types of questions) should be moved to the main paper as they would make it stronger.

**Note From Pc:**

if the above contains the word "oral" or "spotlight" please see: "oral" presentation means -> notable-top-5% and "spotlight" means -> notable-top-25%. As stated in our emails, we are disassociating presentation type from AC recommendations